# Learning alters salience and proactive attentional priority
Dock H. Duncan [1,2] ✉, Dirk van Moorselaar[1,3] & Jan Theeuwes[1,2,4,5,6]

The ability to ignore salient yet irrelevant stimuli is essential to accomplishing even simple tasks. Previous research has shown that observers are better able to suppress distracting stimuli via experience; yet the precise mechanisms of this learned suppression is a subject of debate. The current study (n = 230) employed a psychophysical approach combined with computational modeling to examine how learned spatial suppression affects perception and performance. The results show that items presented at suppressed locations are perceived as less bright than those in non-suppressed areas, suggesting that learned suppression directly affects the perceived saliency of items. To determine how this saliency change affects visual search, a computational modeling approach was used to compare various models of attentional selection. This analysis favored a model in which learned suppression reduces the saliency of objects presented at suppressed locations in the initial salience calculation. Since the saliency of these items is reduced, they are less able to compete for attentional processing and capture attention less often.

The ability to spot a friend in a crowded stadium or find our favorite cereal on a busy supermarket shelf shows the brain's remarkable capacity to filter out irrelevant stimuli and focus in on what is important. This process of filtering the often-chaotic perceptual input is a fundamental function of the brain, forming the basis for all higher cognitive functions. The mechanism known as selective attention[1], allows us to effectively navigate complex environments by attending information that is relevant for our current goals while discarding information that potentially can interfere with the execution of tasks[2,3].

Many theories have debated the mechanism underlying this attentional function, with dominant theories of attention proposing a dichotomy wherein bottom-up salience and top-down goals interact to shape cognitive processing[4–6]. The flow of perceptual input through the visual processing hierarchy is assumed to give rise to a putative "spatial priority map" which is a representation that guides attention allowing the brain to prioritize certain areas of space based on the interaction between the top-down goals and salience of stimuli in the environment[7,8]. Recent research, however, has challenged this classic view, and proposed instead that previous selection episodes may also affect attentional processes above and beyond top-down and bottom-up influences[9]. These experiences (dubbed "selection history") shape attention in dissociable ways from top-down and bottom-up attentional guidance, and thus motivated the revision of the traditional attentional dichotomy into a tripartite model of attention[9–12].

Previous studies have shown that humans are sensitive to spatial regularities in the environment[13]. Experience with these regularities strongly biases visual search performance. For example, research has shown that attention is sharpened at locations likely to contain a target[14–16]. It is assumed that within the spatial priority map, locations that frequently contain a target are upregulated such that attention is biased towards these locations[17]. While prioritized access within the spatial priority map is well understood[18], it is less clear how the brain filters out irrelevant and distracting information. Recent studies have shown that through statistical learning, observers extract spatial regularities regarding salient distractors, resulting in reduced interference with visual search for targets[19–25]. These studies typically introduce spatial regularities by presenting a salient yet irrelevant distractor more frequently at one location than at others. The common finding is that visual search for a target is faster and more accurate when the distractor appears at this high-probability (HP) location compared to other locations[20,23]. To explain this, it has been suggested that, within the assumed spatial priority map, the frequent distractor location becomes suppressed relative to the other locations[13].

While the notion of learned suppression is relatively undisputed[26,27], there is ongoing debate about the mechanisms of suppression within the priority map. Some suggest that suppression is reactive, following the rapid disengagement of attention from the attended object occurring only after the search display has been presented[28,29]. Other researchers argue that

[1]Department of Experimental and Applied Psychology, Vrije Universiteit Amsterdam, Amsterdam, the Netherlands. [2]Institute for Brain and Behavior Amsterdam (iBBA), Amsterdam, the Netherlands. [3]Faculty of Social and Behavioral Sciences, Experimental Psychology Helmholtz Institute, Utrecht University, Utrecht, the Netherlands. [4]William James Center for Research, ISPA-Instituto Universitario, Lisbon, Portugal. [5]Department of Psychology and Behavioral Sciences, Zhejiang University, Zhejiang, China. [6]Mind, Brain and Behavior Research Center (CIMCYC), University of Granada, Granada, Spain. ✉e-mail: D.H.Duncan@vu.nl

suppression occurs proactively, suggesting that it is already established before the display appears[20,30,31]. One prominent hypothesis that is consistent with a proactive account suggests that changes in synaptic excitability within neural networks facilitate the retention of information from past selection episodes, thereby influencing future attentional priorities via a Hebbian-learning mechanism[32–34]. According to this "synaptic theory", attention is modulated in different spatial regions based on previous experiences, either enhancing or suppressing attention. Although the exact nature of this learned attentional bias is still debated, recent electrophysiological work suggests that learning may fundamentally alter the salience of sensory inputs[35]. This refinement of the priority map emphasizes previously relevant locations and de-emphasizes previously distracting locations.

If, through learning, salience processing is impaired at locations within the spatial priority map, it is predicted that any stimulus presented at such a suppressed location will be perceived as subjectively less salient. To test this hypothesis, we combined a visual search task involving learned suppression with a psychophysical approach—a known method of probing perception[36]—and a computational modeling approach. Participants performed a modified version of the additional singleton task with an imbalanced distractor distribution—a paradigm that can be used to implicitly train participants to expect distracting information to appear in certain regions of space[20,22,23]. Critically, the search display was occasionally replaced by a subjective brightness task wherein participants had to select which of two patches was perceived as being brighter (Experiment 1A) or darker (Experiment 1B), while differences in patch brightness was controlled by a staircase procedure. In Experiment 2, we changed the search stimuli such that participants searched among colored shapes rather than black and white shapes to see whether the results observed in Experiments 1A and 1B would generalize to more commonly used stimuli. We paired these findings with computational modeling of the response time distributions in the visual search task to test whether the learning effect was best characterized by a reduction in capture episodes by distractors at the suppressed location, or alternatively by a reduction in capture time by these distractors.

## Methods
### Open data statement
All anonymized participant data, as well as experimental code, analysis scripts and preregistrations for Experiments 1 and 2 are available on the project's PSF repository (https://doi.org/10.17605/OSF.IO/25F76. Experiment 1B was preregistered on 26-12-23 and Experiment 2 was preregistered on 13-1-25). All deviations from preregistration are explicitly stated.

## Experiment 1
### Participants
One hundred and thirty participants (93 men, 37 women self-reported. Median age 28) participated in Experiment 1 split across two subexperiments (40 and 90 participants for Experiments 1A and 1B respectively). The sample size for Experiment 1A was based on an estimate of medium effect sizes in our comparisons of interest (Post hoc sensitivity calculated using a single-sample two-tailed $t$-test indicated 40 participants were sensitive to effects of $d > 0.58$ with an alpha of 0.05 and power of 0.95). The sample size for Experiment 1B was preregistered and based on the observed effect size recorded in Experiment 1A's staircase analysis which yielded an effect size of 0.43 (see "psychophysical modeling results" below). Our sample size of 90 participants thus gave us more than a 99% chance of observing the effect if present in the data[37]. All participants were recruited via the online platform Prolific (www.prolific.co). The experiment was approved by the Ethical Review Committee of the Faculty of Behavioral and Movement Sciences of Vrije Universiteit Amsterdam and adhered to the declaration of Helsinki. All participants indicated their informed consent prior to the experiment. Because social economic statuses, communities of descent and racial/ethnic information were not factors of interest in this study, no such information as collected. Participants accessed the experiment using their personal computers via the experiment hosting website

JATOS[38] after recruitment on Prolific. The experiment was programmed in Javascript using OpenSesame software[39]. Both experiments lasted ~1 h, and participants were compensated £8.50 for their participation. Participants were explicitly informed that their data would be checked for guess behavior before acceptance and payment and would be rejected if the experimenter suspected low effort (i.e., close to random chance performance on the search task). In order to participate in the experiment, prospective participants had to be between the ages of 18 and 42, have a 99% or better acceptance rate on the platform, have participated in at least five other experiments and had have not participated in any other similar experiments from the same research group.

If a participant's accuracy was 2.5 standard deviations (SD's) away from the group average, they were excluded and replaced with a new participant (4 participants replaced). Additionally, to ensure that participants' laptops were able to display the stimuli properly, a post experiment test asked participants to match the brightness of two shapes presented on the screen (see "Methods" below). If a participant reported a salience value for one of the tests that was more than 30 degrees in color space away from true equality, then their data was excluded and a new participant recruited (27 participants replaced). Due to a programming error, the final screen test values were not recorded for an additional 13 participants, who were simply replaced. Finally, if a participant selected the wrong item during the saliency probe on more than 45% of trials, they were excluded and replaced (21 participants replaced).

### Design and procedure
Participants performed a modified version of the additional singleton paradigm[40] with an imbalanced spatial distractor distribution[23]. Occasionally (40% of trials), instead of a search trial, participants instead were presented with a two-item probe trial where a saliency judgment task was performed. Each experimental session started with a comprehensive description of both tasks, including 24 practice trials for the saliency task and 80 practice trials for the search task, presented in separate training blocks. Participants were told at this point the importance of maintaining fixation during the experiment. For the search task, participants were presented with a circle of eight equidistant, evenly spaced items around fixation. Each item could either be a circle or a square. On each array, seven of the items would be the same shape and one item would be a unique shape (randomly selected on each trial). The participants' task was to find this unique shape on every trial. Embedded within each shape in the circular array was either a vertical or horizontal white line (four verticals, four horizontals on each trial randomly distributed in space). Once participants had identified the unique shape in the array, they were to report the orientation of the line embedded within this shape by pressing the "up" or "left" arrow keys for vertical and horizontal lines respectively. On 88% of trials an additional color singleton distractor would be present. While the search items were colored gray (rgb 187,187,187), this singleton distractor was colored black, with a gray outline distinguishing it from the background (see Fig. 1A). We chose to use consistently darker distractors than targets to avoid chromatic adaptation[41]. Participants were given 2000 ms to respond, after which the trial would be marked as incorrect. If an incorrect answer was given, negative feedback was given in the form of a red fixation cross displayed for 400 ms. No positive feedback was given on correct trials as the negative feedback also acted as a time penalty to encourage participants to give correct answers to shorten the overall experiment time.

While targets appeared at each of the eight locations in space with equal probability per experimental block (18 trials per location of 144 trials total per block), distractors appeared disproportionately more often at one location (50% of distractor present trials; 56 trials per block). This HP location was assigned at the beginning of the experiment and could either be the far right or far left location on the horizontal midline, counterbalanced across participants.

Participants completed five experimental blocks consisting of 144 search trials each. Following each trial, a 750 ms intertrial period would occur where only the fixation cross was presented on the screen. The fixation

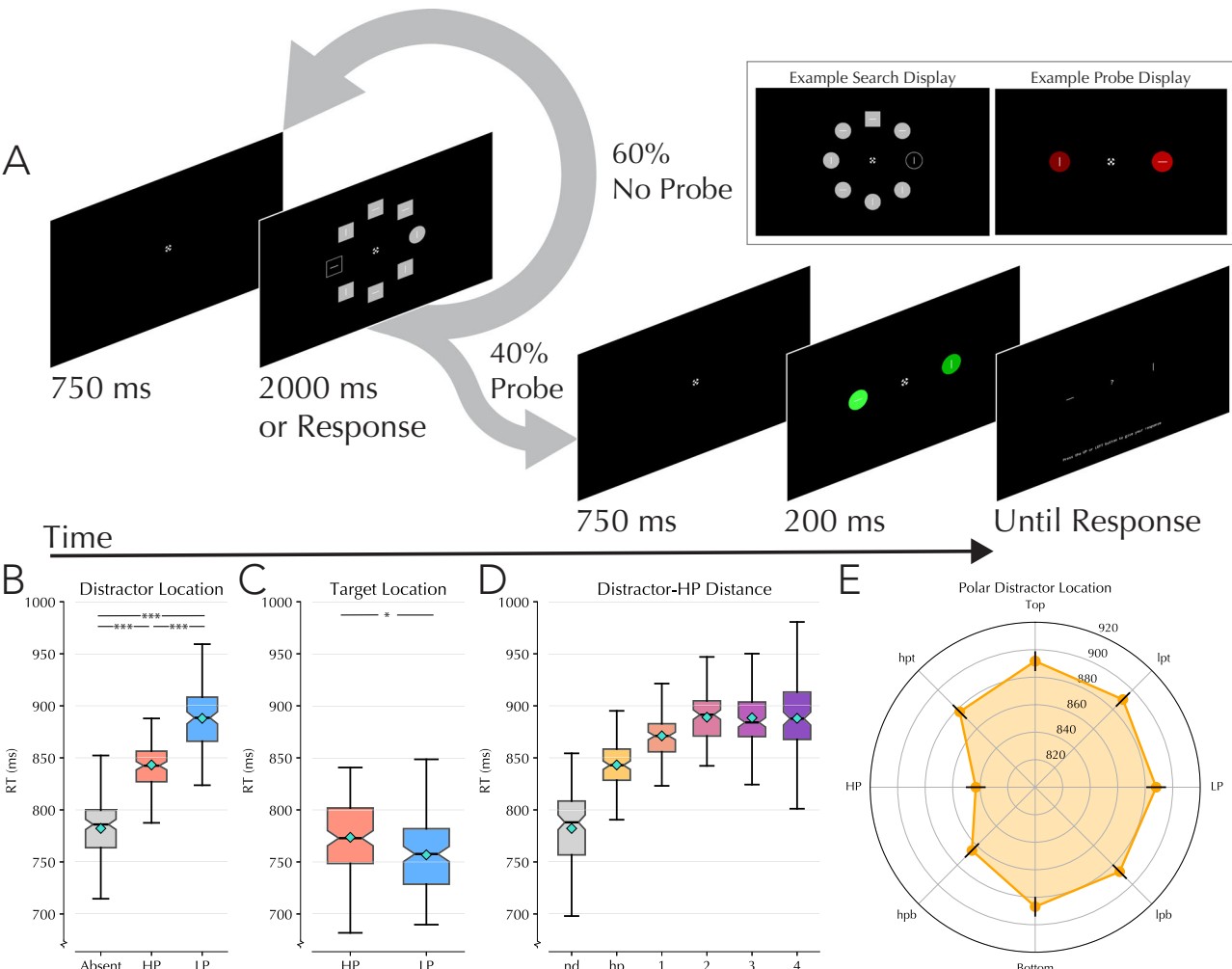

**Fig. 1 | Paradigm and behavioral results for Experiments 1A and 1B.**
**A** Experimental design. Participants performed a visual search task where they searched for a unique shape among an array of eight shapes presented equidistantly from a central fixation point on a black background. In Experiment's 1A and 1B, the target shape was either a gray circle or square, swapping randomly between trials. In 88% of trials, at some location in the array, a black color singleton distractor was present. On a subset of trials (40%) search trials were followed by a probe trial at the moment the participant expected a new search trial to occur. Probe trials consisted of two-colored shapes presented to the left and right of fixation at the locations that the search stimuli could have appeared on the horizontal midline. In Experiments 1A and 1B, these probe shapes were either both colored green or both red, randomly selected on each trial. Embedded in each probe shape was a white vertical or horizontal line, which would remain on screen after the stimuli themselves disappeared. Participants were asked on probe trials to indicate which side held a brighter stimulus (or, in Experiment 1B, which contained the darker stimulus) by selecting the line at the location this stimulus was presented using the same response coding scheme as on search trials. Text on the bottom of the probe response screen reads "press the up or left button to give your response." **B** Average response times for trials in which a distractor was absent, for when the distractor was presented at the high-probability location, or for when presented at a low-probability location in

Experiments 1A and 1B ($n = 130$). Note that these analyses are restricted to trials in which the distractor was at one of the horizontal locations. **C** Average response times for trials in which the target was presented at the high-probability distractor location, or a low-probability distractor location. Note that these analyses are restricted to trials in which the target was at one of the horizontal locations and no distractor was present. **D** Gradient of suppression relative to the HP location from Experiments 1A and 1B data. Nd indicates trials in which no distractor was present, HP is when the distractor was at the high-probability location, 1 indicates the distractor was immediately adjacent (above or below) the high-probability location, 2, 3, and 4 represent how far away the distractor was from the high-probability location (with 4 representing the opposite side of the screen). For pairwise statistics, see Supplementary Table 1. **E** Radial diagram of distractor position interference relative to the high-probability location. All participant data from Experiments 1A and 1B have been collapsed and aligned such that the high-probability location can be represented at the same location (the left-most location). For a full breakdown of the results of comparing location-based suppression, see Supplementary Table 2. All boxplots reflect within subject variance[114–116]; shaded box extends over IQR; the middle line represents median; whiskers extend to mini/maximum values; turquoise diamonds indicate condition means; notches indicate 95% confidence interval of median. *** = $p < 0.001$; ** = $p < 0.01$; * = $p < 0.05$

cross used was adapted from Thaler et al.[42] to encourage fixation. Following every search trial, there was a 40% chance that rather than another search trial, instead participants would next perform a probe discrimination task. On probe trials, rather than an array of eight shapes, instead only two shapes would appear. These shapes were presented on the horizontal midline at the same locations as the outer left and right items within the search array. Due to a programming error, in Experiment 1A these two shapes were always circles. This error was corrected in Experiment 1B, where they could be

either two circles or two squares. The shapes were presented in red or green, with the colors always matching (either two red or two green shapes were presented). Supplementary Fig. 1 visualizes the color spectra from which the coloring of these shapes could be selected. On one end of the spectra was black, and on the other white, with 509 intermediary shades of green or red. On every trial, one of the shapes would be a standard color—corresponding to position 187 in the color spectra (highlighted as the black line in Supplementary Fig. 1). The other color would be different, and the participants'

task was to perform a brightness discrimination task on these two shapes. In Experiment 1A, the participants' task was to select the brighter shape. To account for potential response biases relative to the HP distractor location, in Experiment 1B then the participants' task was changed to selecting the darker of the two shapes. Shapes were only presented on the screen for 200 ms, after which they disappeared. Embedded in each shape was a white line which would remain on screen after the shapes had offset. The lines could either be oriented vertically or horizontally (one each on every trial) and participants provided their answer by selecting the line orientation present at the location that they believed the brighter (or in Experiment 1B, darker) shape had been presented.

The standard colored shape could be presented on either the right or left side of the screen, randomly selected on each trial. Critically, the brightness of the nonstandard color item present on the opposite side was controlled by way of an adaptive staircase procedure meant to home in on the point of subjective equality by identifying the location where participants were equally likely to select the standard item or the staircased item (50% likelihood point). The staircasing method was based on the interval bisection method used in root finding, and was essentially a simplified version of the "parameter estimation by sequential testing" (PEST) method[43,44]. The critical insight of the PEST method is that not all datapoints are equally informative, and that maximizing the number of the most informative datapoints in a dataset can drastically reduce sampling needs without sacrifices to signal quality[45]. In the current case - as the point of subjective equality is the main research interest - the point around where participants are equally likely to select either of the two-colored shapes will produce the most informative datapoints, while extreme color values in which participants virtually always select the correct colored shape are less informative. To design an adaptive staircase that homed in on the case the 50% guess point, we used a custom algorithm using collapsing step sizes following consecutive response alternations. After initializing a new staircase, the starting point would be placed far from true equality (randomly either 28 or 350, set separately for HP and LP locations) and step size would be set at the standard value of 64 degrees in color space. What this meant is that following each probe trial, if a participant selected the non-standard color, then on the next trial in which that side was tested then the color would be 64 degrees brighter in hex space than on the previous iteration (or darker in Experiment 1B; see Supplementary Fig. 1 for visualization of full 509 degree color spectra). On each probe trial, it was recorded whether the participant selected the same or the different side relative to the last probe trial. When participants selected a different side three trials in a row, then the step size was reduced by half. This was done because presumably participants would vacillate around the 50% point. Over time, as participant completed trials, the step size would become smaller and coalesce around the 50% point. This was done separately for the left and right side of the screen.

At the end of each experimental block, the staircases for both sides were reset such that at beginning of subsequent blocks, the initial color value would be far away from the 50% point and step sizes would be large (64 degrees). This was done as we expected the learning effect to develop over the experimental blocks, but staircases become more restricted as more trials pass and the step size shrinks. By resetting the staircase, we allowed it more flexibility to identify shifting 50% points. Furthermore, this allowed the participant to encounter more easy trials, discouraging them from simply guessing on each trial. The starting point on each new block was calculated as 96 degrees away from the staircase ending point of the previous block, and could randomly be either brighter or darker. This ensured that ratings on the next block would be centered around the current best guess for the actual point of subjective equality.

Following the completion of the experiment, participants were told that one location was more likely to contain the distractor than any other location and were asked if they had noticed this regularity during the experiment. Regardless of their answer, all participants were next asked to guess where they thought distractors were most likely to appear by selecting a number corresponding to the eight locations in an example screen. If a participant both indicated that they knew the regularity and also selected the

correct location for their condition, then they were marked as aware of the experimental manipulation. After probing participants' knowledge of the experimental manipulation, next they completed a screen test in which the brightness on the left and right side were matched. This was done in two steps. First two red squares were shown on the left and right side of fixation at the exact locations that the probe items had previously appeared in. Using the vertical arrow buttons, the participants were able to change the brightness of the square on the left side of the screen by moving it up and down the color spectrum. Participants were asked to match the brightness of the two items while both remained displayed on the screen. Next, participants were given the same task but this time two green squares were displayed, and participants controlled the brightness of the stimuli on the right side of the screen. On both screens, the neutral stimuli were the standard color used during the experimental session, thereby allowing us to measure whether they were able to match the saliency of the two items when given unlimited time for comparison.

### Behavioral preprocessing

Search trials in which the response time was less than 300 ms or longer than 2000 ms were excluded from analysis (1.1% of all data). Additionally, participant mean RTs and standard deviations were calculated and trials more than 2.5 SDs away from participant averages were excluded (2.5% of data). In all analyses except accuracy analyses, incorrect trials were also excluded (7.2% of data). For analyses on distractor locations, distractor repetitions were excluded to avoid well known intertrial effects[22] (19.5% of data). In target based analyses, target repetitions were similarly excluded for target based intertrial effects[14,46] (11.2% of data). Furthermore, in the main analysis of distractor presence at HP and LP locations, trials were controlled such that only trials in which the distractor was on the horizontal midline were included. Items presented horizontally benefit from the well-known vertical meridian asymmetry effect[47] whereby items presented horizontally are better processed than those presented vertically. This could have the effect of artificially inflating our statistical learning effect as we would be comparing distractors presented horizontally (HP) to those presented both horizontally and vertically (LP). This same subset of trials was also used for fitting our modeling results. For the main target-based analysis, trials were restricted to those in which no distractor was present, and targets were presented on the horizontal midline for the same reasons as discussed above.

### Statistical tests

Our analyses will rely on simple repeated measure analyses of variance (rmANOVA) or *t*-tests. Ninety-five percent confidence intervals (CI) reported for *t*-tests indicate the interval of the reaction time difference between conditions. In all cases assumptions of sphericity (for ANOVAs) or regularity (for *t*-tests) will be conducted and, if violated, the proper corrected statistical test will be performed. In the case of rmANOVAs, Greenhouse-Geisser corrected ANOVAs (ggANOVA) will be conducted, and for *t*-tests, Wilcoxon signed-rank test (*w*-test) will be performed instead. As can be seen in the reaction time modeling data, our results generally did not follow a normal distribution, thus motivating a nonparametric test. Rather than normalizing the data (e.g., via log transformation), we chose to preserve reaction time information and defer to appropriate non-parametric tests.

### Staircase analysis

Because the staircase reset at the beginning of each experimental block, the main points of interest were staircase values at block ending points. Each participant produced five datapoints of interest (for the five blocks) which were averaged to make a combined staircase ending score for the combined staircase analysis.

### Psychometric curve analysis

Participants' perceptual data was analyzed by creating a "combined salience" score. This score calculated the relative difference between the probe presented at the HP location versus the LP location on every probe trial, such that positive values were a more salient HP location probe and negative

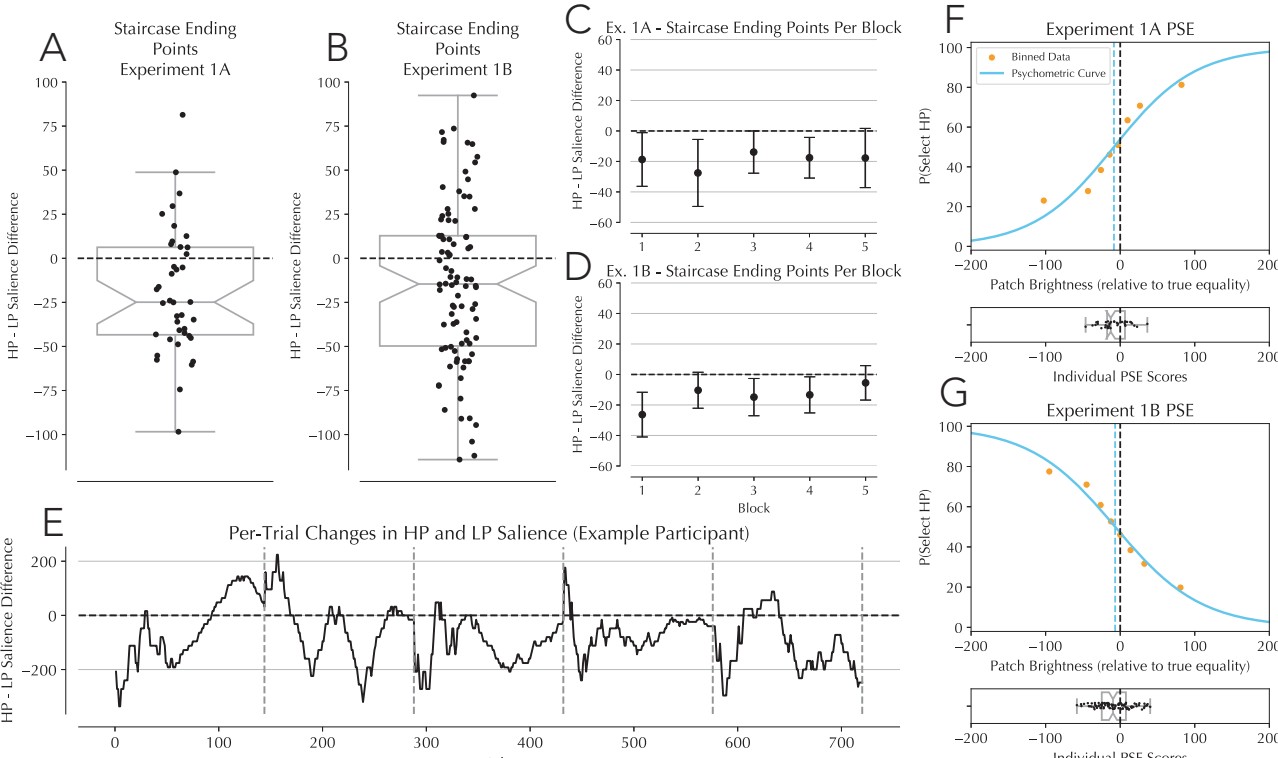

**Fig. 2 | Staircase and psychophysical modeling results. A** Staircase ending point differences from Experiment 1A ($n = 40$) in which participants needed to select the brighter of the two probe stimuli. Dots represent each participants' staircase ending point averaged over the five blocks. Difference was calculated by subtracting the HP ending point from the LP ending point per block. **B** Staircase ending points differences from Experiment 1B ($n = 90$) where participants needed to select the darker of the two probe stimuli. **C, D** Per-block average staircase ending point in Experiment 1A (**C**) and 1B (**D**). **E** Here is shown an example participant's (from Experiment 1A) per-trial staircase behavior. **F** PSE calculated for aggregate and individual participant data in Experiment 1A. Top panel: aggregate participant data separated into eight bins of equal sized based on the difference in patch brightness. Orange dots represent the bin's mean probability to select the HP location. the placement of the

orange dots on the x-axis was determined by taking the mean patch brightness score within each bin. PSE was calculated by finding the midpoint of a logistic function fitted to the arranged binned data. The vertical blue line marks the PSE for the aggregate analysis. Lower panel: PSE scores when restricting the binning and curve fitting procedure to individual participants' data. **G** PSE calculated for aggregate and individual participant data in Experiment 1B. Binning and curve fitting procedures mirrored those used in Experiment 1A. Note that the curves in A and B are mirrored as the participant had opposite tasks—so a shape that was much darker than true equality in Experiment 1 should reliably not have been selected, while the opposite was true in Experiment 1B. The vertical blue line marks the PSE for the aggregate analysis.

items were a less salient one. Participant responses were then coded as either selecting the HP probe or not. Because the staircase procedure ensured the density of the combined salience values would cluster around the PSE, and because ultimately participants rated items at HP locations as less bright than at LP locations, the overall distribution of combined salience values was centralized below zero with relatively few trials occupying extreme positive and negative values. To fit a psychometric curve to this data, the data was separated into eight bins of different combined salience scores with each bin accounting for the same number of trials (1/8 of total data). Bin labels were defined as the mean combined salience value within each bin. P(hp) was then calculated as the probability of selecting the HP location, and bins were plotted at the intersection of the bin label and the bins P(hp) value. A psychometric curve was then plotted onto these datapoints with a starting midpoint of 0 and slope of 0.01 (-0.01 for Experiment 1B). This curve fitting procedure was done both on the aggregate participant data (Fig. 2F, G, top panels), as well as on a per-subject basis (Fig. 2F, G, bottom panels). The PSE was then calculated for each participant and the PSE distribution against zero was tested to determine whether participants reliably rated items as less bright at HP locations relative to LP locations.

**Reaction time distribution modeling**

In an unregistered additional analysis, we sought to investigate whether the shape of the reaction time distribution in distractor present and absent trials could be used to draw inferences about the nature of the mechanism

underlying learned suppression using computational model comparisons. Our model of RT data was based on the simple assumption that search trials should by-and-large consist of two trial types: trials in which attention went directly to the target, and trials in which attention was first captured by the distractor followed by disengagement and re-capture by the target singleton[48–50] (Fig. 3B). The overall RT distribution should then represent the sum of two curves. The shape of the no-capture curve was modeled by using no-distractor trials where it is reasonable to assume that attention reliably went directly to targets. Distractor present trials were then constructed as a mix of the no-capture curve with another capture-curve. To begin, we tested several different curves to find which one matched the distribution of our no-distractor trials best. Specifically, we tested a log-normal, Weibull, chi-squared, inverse gamma and an exponential-gaussian curve to fit the data (Supplementary Fig. 2). A comparison of BIC scores across curve fits clearly identified the ex-gaussian distribution as the best fit to our RT data, and so we proceeded to fit our distractor present trials using ex-gaussian curves (note that this accords with recent work modeling RT distributions)[51–54]. For our model comparison, three different models were constructed to account for differences in the RT distribution between HP and LP trials. The first was the capture model, where the number of capture episodes by distractors was varied between HP and LP distractor trials. This was captured in our model by varying the size of the curves before summation by multiplying our no-capture curve by some variable between zero and one (x) and our capture curve by one minus this variable (1 − x). Our

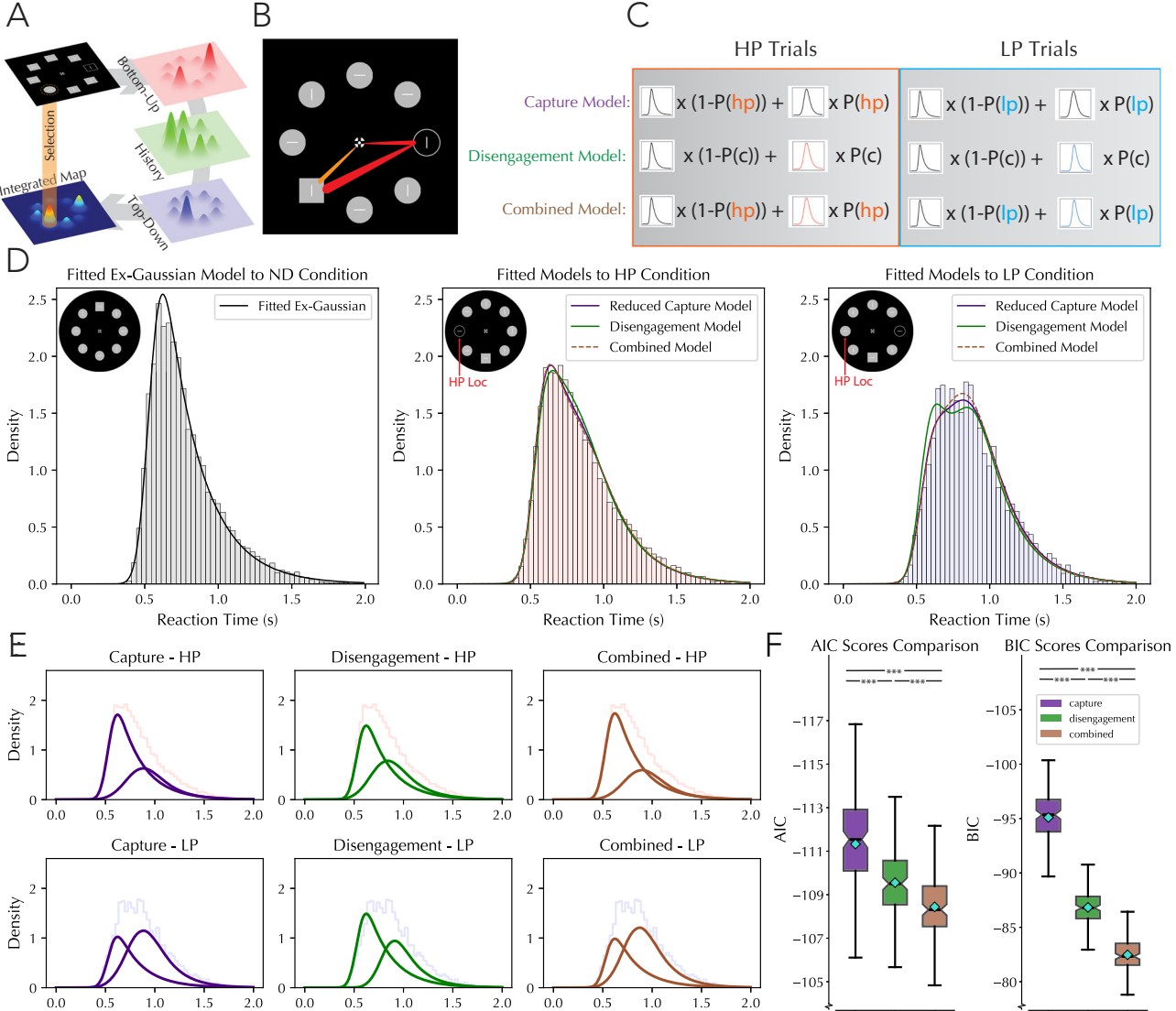

**Fig. 3 | Computational modeling of RT distributions. A** graphic of the hypothesized interaction between the three components of attention (top-down attention, bottom-up attention and selection history). Perceptual input is thought to pass through these three layers and produce a integrated priority map from which attentional selection is made based on the region with the highest calculated attentional priority. **B** Illustration of hypothesized attentional orienting on capture trials (red) and no-capture trials (orange). **C** Illustration of the three RT models compared in the current analysis. The capture model characterizes the difference between HP and LP distractor trials as mediated by a different rate of capture by distractors. This is represented by unique capture odds (P(hp) and P(lp)) but the same curve for each trial type. The disengagement model characterizes the difference between HP and LP trials not as differences in the rate of capture, but differences in post capture dynamics. This is represented by unique curves (colored red and blue) but a constant rate of capture (P(c)). The combined model allows both the capture curve and the rate of capture to vary between HP and LP trials. **D** Outcomes of curve fitting on no-distractor, HP-distractor and LP-distractor trials superimposed over the aggregate RT histograms. For no-distractor trials, a baseline curve was fitted representing how attention moves directly to targets when no competition from a distractor is present. This no-distractor curve was then used in the two distractor-present trials. The presumption was that some trials would follow this no-distractor curves distribution—as on some trials participants would simply directly attend to the target without any distractor processing. This subset would then be blended in

with another population of trials to get the overall RT distribution. For the two distractor present trial populations, the fitted curves for our three models are represented as different colored lines. Note that all three trials do a relatively good job at characterizing the data, with the differences coming mostly in the margins. The AIC scores for the capture, disengagement and combined models were −661, −641 and −657, respectively. The BIC scores were −626, −592 and −601, respectively In all tested conditions, trial selection was restricted to those in which the distractor was on the horizontal midline (or absent). **E** Separated model curves before addition. The model curves in (**D**) were the addition of two constituent curves, these curves are shown separately in this figure for each model and each distractor condition. Note that the model components are identifiable in these curves; with the capture model using the same shaped curves in HP and LP condition but varying their sizes, while the disengagement model used the same sized curves for HP and LP conditions but varied their shapes. **F** Average AIC and BIC scores for each of the three models when fit to individual participant data. All boxplots reflect within subject variance[114–116]. *** = $p < 0.001$. Pairwise comparisons: AIC, capture vs, disengagement ($W = 2269$, $p < 0.001$, r_rb = 0.73, [CI: −2.67, −1.23]); capture vs combined ($W = 1451$, $p < 0.001$, r_rb = 0.83, [CI: −3.62, −2.36]); combined vs. disengagement ($W = 1510$, $p < 0.001$, r_rb = 0.82, [CI: 0.96, 1.75]). BIC, capture vs disengagement ($W = 127$, $p < 0.001$, r_rb = 0.99, [CI: −9.165, −7.720]), capture vs combined ($W = 0$, $p < 0.001$, r_rb = 1.00, [CI: −13.37, −12.10]), combined vs disengagement ($W = 220$, $p < 0.001$, r_rb = 0.97, [CI: 4.196, 5.025]).

second model was a disengagement model, were the odds of capture by a distractor was kept constant between HP and LP distractor trials but the curve shape itself varied. This allowed the model to represent the dynamics of disengagement differently between the models by, for instance, varying the midpoint of the two distractor capture curves. The third model was a combined model which allowed both the curve and the capture odds to vary between HP and LP conditions but penalized the extra model complexity. Our ex-gaussian curves included three variables, K, Loc and scale corresponding the curve's skew, midpoint and horizontal spread respectively. The formula for the exponential-normal distribution is as follows:

$$f(y, K) = \frac{1}{2k} \exp\left(\frac{1}{2k^2} - \frac{y}{K}\right) erfc\left(-\frac{y - \frac{1}{k}}{\sqrt{2}}\right) \qquad (1)$$

Where:

$$y = \frac{x - loc}{scale} \qquad (2)$$

And "erfc" represents the complementary error function. Importantly, the reduced capture and combined model offered the possibility that capture odds could be set at 100%, meaning that the curve fit to the no-distractor trial could be entirely discarded. To prevent this, capture was set at a floor of 20% and a ceiling of 80%, ensuring that the no-distractor curve would be continuously influential (Note that exploratory analyses which allowed this variable to range up to 100% or down to 0% produced roughly the same results). For model comparisons, our capture model was treated as having five variables (k, loc, scale, capture_hp, capture_lp), our disengagement models as having seven (k_hp, loc_hp, scale_hp, k_lp, loc_lp, scale_lp, capture) and the combined model as having eight (k_hp, loc_hp, scale_hp, k_lp, loc_lp, scale_lp, capture_hp, capture_lp). In exploratory analyses, the capture and disengagement models were also compared while assuming equal parameter numbers, and it was found that this did not lead to a reversal of the results. These model parameters are compared in Fig. 3C, and their fitted curves visualized in Fig. 3D, E.

## Experiment 2
### Participants
A preregistered power analysis based on the observed effect size in our psychometric PSE analysis from Experiment 1B ($d = 0.335$) suggested that 100 participants would have a 95% likelihood of observing a significant result using one-tailed one-sample *t*-tests on our variables of interest (individual PSE scores). Based on this, 100 participants (75 men, 25 women self-reported. Median age 27) participated in Experiment 2. All consent forms and ethical approval as well as participant recruitment and selection criteria were the exact same as in Experiments 1A and 1B. Participants were again compensated £8.50 for their participation.

4 participants were replaced as their accuracy was 2.5 SD's away from the group average. A further 14 were replaced for failing the post hoc screen test, and 29 were replaced because their accuracy was less than 55% on the salience probe trials.

### Design and procedure
The experimental design of Experiment 2 exactly mirrored that of Experiment 1A except that participants now saw search arrays that were colored either red (187,0,0 rgb) or green (0,187,0 rgb; See Fig. 4A). Probe trials were conversely changed to gray (Fig. 4B), with the standard color set to position 100 in the gray color space (100,100,100 rgb; see Supplementary Fig. 1). Distractors were again present on 88% of trials and would always be the non-target color (i.e., if the shapes were green, the distractor was red, and vice versa). Additionally, and similar to Experiment 1B, the programming error was corrected such that probes could now be squares or circles.

## Results
### Experiment 1
**Behavioral analyses.** Given that the search task did not differ between Experiments 1A and 1B, all subsequent analyses of search times were collapsed across Experiments 1A and 1B. As visualized in Fig. 1B, an rmANOVA with within subjects' factor Distractor condition (No-distractor, high-probability distractor and low-probability distractor) confirmed that distractor interference was modulated by the uneven distribution of distractors (ggANOVA: $F(1.64, 211.6) = 138.7$, $p < 0.001$, $\eta^2 = 0.518$, [CI: 0.435, 0.602]) Critically, planned pairwise comparisons showed reduced interference by the salient distractor when it was presented at the HP compared to the low-probability (LP) locations (*w*-test: $W = 2181$, $p < 0.001$, r_rb = 0.744 [CI: −47.68, −21.57]). As in previous work[20,23], this spatial suppression was not restricted to distractor processing, as within distractor absent displays, target processing was impaired at HP relative to low-probability locations (*w*-test: $W = 3384$, $p = 0.042$, r_rb = 0.603 [CI: 0.672, 28.1]; Fig. 1C).

Previously, it has been demonstrated that suppression effects extend from the HP distractor location to adjacent spatial areas, forming a suppression gradient around the HP location[23,55,56]. Consistent with these previous findings, a previously unregistered analysis revealed significant effects of distractor location relative to the HP location (ggANOVA: $F(3.51, 452.8) = 55.63$, $p < 0.001$, $\eta^2 = 0.301$, [CI: 0.2359, 0.367]; Fig. 1D). Subsequent post-hoc testing showed that this gradient was rather short in scope, with distractors positioned immediately adjacent to HP locations showing reliably faster reaction times (RTs) than distractors 2, 3 and 4 positions away, but no significant differences between conditions were found beyond this (see Supplementary Table 1 for pairwise comparisons). When conducting this suppression gradient analysis while separating upper and lower locations, the same general pattern of results is observed (Fig. 1E; For the complete breakdown of the comparisons between these locations, see Supplementary Table 2).

Finally, we examined whether participants had developed explicit knowledge about the spatial distractor imbalance. Despite robust behavioral effects, participants typically cannot identify the underlying regularity that occurs in these type of tasks[14,23,57], but see ref. 58. This aligns with extensive experimental evidence showing that participants cannot willfully suppress a region in anticipation of a distractor, but can learn to suppress regions or distractor features through experience[59–62]. To assess awareness, we administered a post-experiment questionnaire asking participants whether they noticed the distractor regularity and challenging them to select the HP distractor location. Only participants that both report awareness and were able to select the correct location were then labeled as aware. Only 11 out of 130 participants met these criteria for awareness (a further 26 reported noticing the regularity but were unable to indicate where it was in space and 27 reported not noticing the regularity but selected the correct location anyway). Importantly, the exclusion of aware participants did not abolish the main effect of distractor position (HP vs. LP [$n = 119$]: $W = 1859$, $p < 0.001$, r_rb = 0.74, [CI: −48.27, −20.78]), consistent with an implicit learning mechanism.

**Psychophysical modeling.** It is noteworthy that recent work has used staircased psychophysical approaches to study the interaction of salience and perceptual encoding time[63,64]. These studies have fluctuated the presentation time of stimuli of varying saliences to measure the minimal encoding time needed for their perception. The current approach was essentially an inversion of this approach: keeping the presentation time the same (200 ms) but varying the salience instead. An initial analysis of staircase ending points was done by averaging per participant where the staircase ended on each of the six blocks. This analysis revealed that participants reliably rated the suppressed location as less bright relative to the non-suppressed location ($t(39) = 3.293$, $p = 0.002$, $d = 0.521$, [CI: −30.53, −7.647]; $t(89) = 2.87$, $p = 0.005$, $d = 0.303$, [CI: −23.81, −4.33] for Experiments 1A and 1B respectively; Fig. 2A, B). A subsequent rmANOVA with within subjects' factor Block (1-5) and Experiment as a

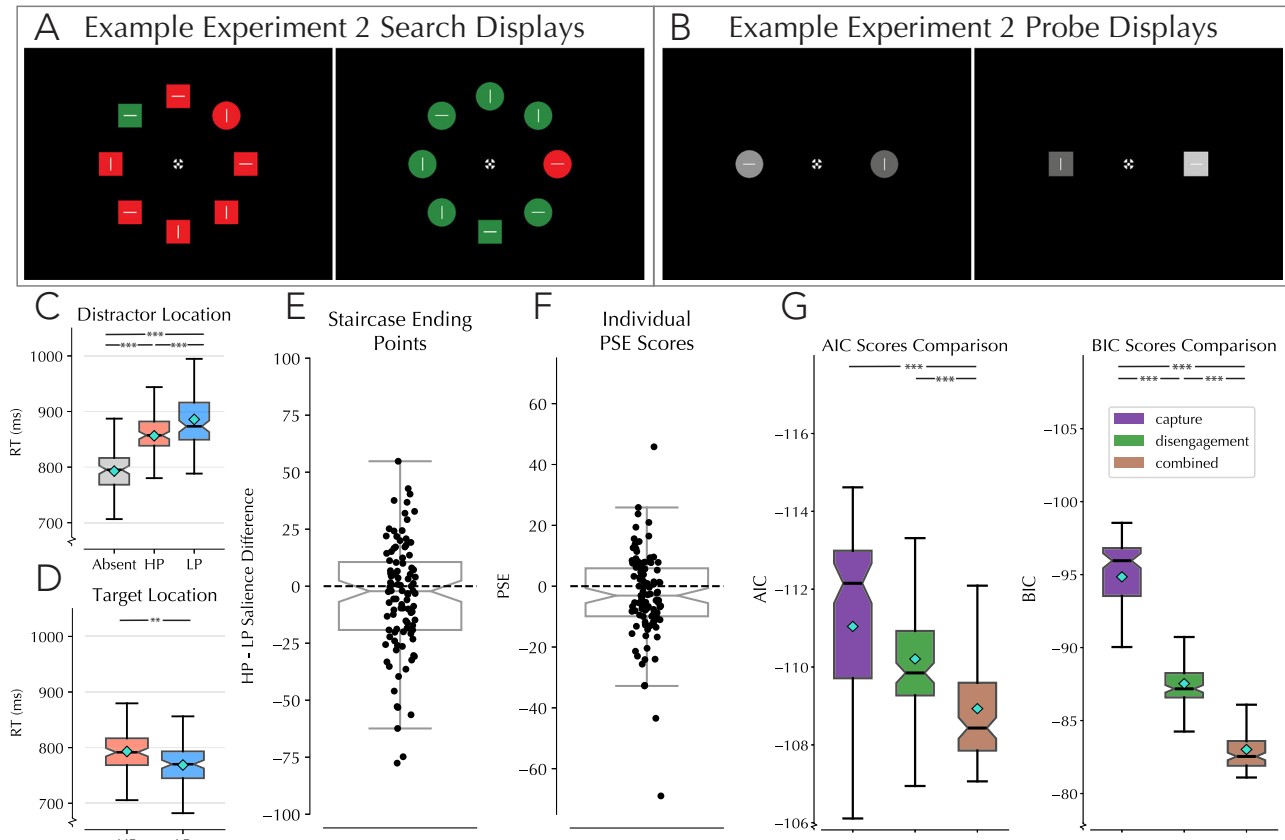

**Fig. 4 | Results of Experiment 2 in which search arrays were either red or green and salience probes were shades of gray.** **A** Example of the colored search arrays used in Experiment 2. Target and distractor colors swapped randomly from trial to trial. **B**) Example of the gray probes used in Experiment 2. **C**) Average response times for trials in which a distractor was absent, for when the distractor was presented at the high- or low-probability location ($n = 100$). Note that these trials were restricted to trials in which the distractor was at one of the horizontal (see "Methods", Behavioral Preprocessing) **D** Average response times for trials in which the target was presented at the high-probability distractor location, or a low-probability distractor location. Note that these trials are restricted to trials in which the target was at one of the horizontal locations no distractor was present **E** Staircase ending point differences from Experiment 2 using gray probes. Dots represent each participants' staircase ending point averaged over the five blocks. **F** PSE scores when restricting the binning and curve fitting procedure to individual participants' data in

Experiment 2. Negative values indicate the suppressed location needed to be brighter to be rated as equally salient to the unsuppressed location. **G** Average AIC and BIC scores for each of the three models when fit to individual participant data. All boxplots reflect within subject variance[114–116]; shaded box extends over IQR; the middle line represents median; whiskers extend to mini/maximum values; turquoise diamonds indicate condition means; notches indicate 95% confidence interval of median. *** = $p < 0.001$; ** = $p < 0.01$; * = $p < 0.05$. Pairwise comparisons: AIC, capture vs disengagement ($W = 1441$, $p < 0.001$, r_rb = 0.71, [CI: −2.27, −0.71]); capture vs combined ($W = 1058$, $p < 0.001$, r_rb = 0.79, [CI: −3.60, −2.01]); combined vs disengagement ($W = 660$, $p < 0.001$, r_rb = 0.87, [CI: 1.22, 1.76]). BIC, capture vs disengagement ($W = 193$, $p < 0.001$, r_rb = 0.96, [CI: −8.73, −7.17]); capture vs combined ($W = 104$, $p < 0.001$, r_rb = 0.98, [CI: −13.27, −11.70]); combined vs disengagement ($W = 30$, $p < 0.001$, r_rb = 0.99, [CI: 4.46, 5.01]).

between-subjects factor failed to find a reliable difference between blocks (ggANOVA: $F(3.56, 459.1) = 1.829$, $p = 0.13$, $\eta^2 = 0.014$, [CI: 0.000, 0.034]; Fig. 2C, D). The suppression effect rather was evident from the first experimental block and remained relatively stable throughout the experiment. Further support for this perceptual change came from our analysis of the point of subjective equality (PSE), calculated by fitting a psychometric curve onto the combined participant data. Consistent with a shift in subjective salience judgment, the PSE was calculated as below zero in both experiments, indicating participants subjectively rated items presented in the suppressed region as less bright (Fig. 2F, G, top panel; for details see "Methods" section). By fitting psychometric curves to individual participant data, we found that the PSE was reliably below zero in both experiments (t-test: $t(39) = 2.82$, $p = 0.008$, $d = 0.446$, [CI: −13.74, −2.264]; $t(89) = 3.176$, $p = 0.002$, $d = 0.335$, [CI: −12.58, −2.897] for Experiments 1A and 1B respectively; Fig. 2F, G bottom panels). This indicates that items at the suppressed location needed to be physically brighter to be perceived as equally salient to those at non-suppressed location (or vice versa).

**Computational modeling.** Learned suppression is widely thought of as implemented on a spatial priority map where top-down, bottom-up and selection history factors converge to make a unified representation of attentional priority across space (Fig. 3A). A key open debate in the field is whether learning effects impose themselves proactively (on the first feedforward sweep of visual processing of a scene) or reactively (requiring some feedback from higher cognitive areas). These competing theories can be thought of as differing in their predictions about the frequency of attentional capture by distractors, with the proactive hypothesis predicting fewer capture episodes by a suppressed distractor while the reactive hypothesis predicts the same number of capture episodes but faster disengagement at suppressed locations[26,29,31,65,66]. While both mechanisms may play a role, the current psychophysical findings make a clear prediction: reduced saliency at HP distractor locations should result in a reduction of capture episodes by salient distractors at those locations relative to non-suppressed locations. For the alternative—faster disengagement—one does not predict a reduction in capture episodes. To test this, we employed a computational modeling approach.

Central to our modeling approach is the assumption that response time distributions in visual search tasks follow a bimodal pattern, representing a mixture of capture trials and trials wherein attention immediately moved towards the target (See Fig. 3B for a visualization of these two trial types). First, we determined the characteristics of no-capture trials by fitting the response time distribution using only trials in which no distractor was present (Fig. 3D, far left). We presumed that these trials accurately represented the response time distribution when attention moved directly to the target. Using this initial fit, three separate models were then created which each blended the distribution from our no-capture trials (calculated from our no distractor trials) with curves representing trials in which attention was first captured by a distractor and then moved to the target. The overall response time distribution on distractor present trials was then taken to be a simple addition between the curve when no distractor capture occurred and new curves which represented trials in which attention was captured by a distractor, with each curve multiplied by a fraction representing the likelihood of capture by a distractor (Fig. 3C). Importantly, HP and LP trials were fitted separately with certain shared variables. Specifically, within one model - labeled the reduced capture model - the proportion of capture vs no-capture trials was allowed to vary between HP and LP trials. By contrast, in the second model - labeled the disengagement model - the proportion of capture trials was held constant in both conditions, but the other characteristics of the capture trial curves was allowed to vary (K, loc. and scale; see "Methods" for details). We reasoned that if learned suppression primarily resulted in faster disengagement, then this should be captured by allowing the curve itself to change in shape and midpoint. Finally, in a third model both capture rate and the other curve characteristics were allowed to vary between the two distractor conditions (see Fig. 3C for summary of these three models).

The reduced capture model outperformed both the disengagement and combined models based on AIC and BIC scores (Fig. 3D). This suggests that the primary effect of attentional suppression is a reduction in initial capture episodes by distractors at suppressed locations, rather than faster disengagement. Specifically, the model estimated 43.6% capture odds by a distractor at HP locations versus 70% at LP locations, while the time course of disengagement remained similar across locations (Fig. 3E). Additionally, our modeled response time curves provided information on the median value on capture and non-capture trials, indicating a rough estimate of how long it would take to recover from singleton capture. Both models used a no-capture curve with a median reaction time value at 714 ms. The reduced capture model further estimated a median reaction time on capture trials of 929 ms, or a total distractor cost of 215 ms. Note that this distractor cost is roughly twice the difference between distractor absent and low-probability distractor trial reaction times, indicating that the presence of no-capture trials in the aggregate reaction times obscures the true temporal cost of visual distraction.

To account for individual differences, we also fitted these models to each participant's data separately. An rmANOVA taking model (capture, disengagement, combined) as factors and AIC/BIC score as the dependent variable corroborated our aggregate findings, with the reduced capture model consistently outperforming the alternatives across individual participants (AIC: $F_{(2,258)} = 35.56$, $p < 0.001$, $\eta^2 = 0.2161$, [CI: 0.1285, 0.304]; BIC: $F_{(2,258)} = 688.6$, $p < 0.001$, $\eta^2 = 0.842$, [CI: 0.808, 0.877]; pairwise comparisons shown in Fig. 3F). These modeling results align with and extend our psychophysical findings, supporting a proactive model of attentional suppression. The reduced capture at suppressed locations suggests that salience signals are modulated prior to the perception of the search array, affecting the initial feedforward sweep of information processing. This proactive modulation results in reduced competitiveness of suppressed regions on the spatial priority map.

## Experiment 2

While Experiments 1A and 1B showed clear evidence for a proactive, saliency-based mechanism of learned suppression, several features of the experimental design cast this result into doubt. Specifically, in Experiment 1,

the distractor singleton was consistently the darker item in the array. Rather than learning to suppress the location of the distractor singleton, as we suggest here, it is possible that observers instead simply learned to expect darker items there. Based on this reasoning, the reduced salience observed for probe items in Experiments 1A and 1B may reflect participants' reliance on a global prior rather than an actual reduction in the perceived saliency of items presented at suppressed locations. This interpretation is consistent with evidence showing that under conditions of increased ambiguity, participants tend to rely more heavily on learned priors[67–69].

Therefore, to account for the potential influence of perceptual priors as well as to bring the current results in line with more commonly used multicolored search arrays, Experiment 2 replicated the design of Experiments 1A, but inverted the color features of the search task and the probe task. Participants now searched arrays of green items with a red distractor singleton, or red items with a green distractor singleton following the classic additional singleton task[40,70]. Salience probes were conversely changed to be shades of gray (Fig. 4A, B).

**Behavioral analyses.** As expected, an rmANOVA with distractor condition as factors found a reliable effect (ggANOVA: $F_{(1.65,163.4)} = 71.17$, $p < 0.001$, $\eta^2 = 0.4182$, [CI: 0.316, 0.521]; Fig. 4C), with HP distractor trials being reliably faster than LP distractor trials (w-test: $W = 1742$, $p = 0.007$, $r\_rb = 0.655$, [CI: −41.07, −5.514]). Furthermore, target processing was impaired at this location, indicating global suppression ($t_{(99)} = 3.0$, $p = 0.004$, $d = 0.298$, [CI: 8.149, 40.56]; Fig. 4D). The suppression gradient analysis additionally revealed that suppression extended beyond the frequent distractor location itself, affecting adjacent locations as well. (see Supplementary Table 2 for all pair-wise comparisons)

Of the 100 participants, 8 reported being aware of the regularity and were able to select the correct HP location (a further 33 reported noticing the regularity but were unable to indicate where it was in space and 20 reported not noticing the regularity but selected the correct location anyway). Again, the exclusion of aware participants did not abolish the main distractor effect (HP vs. LP [$n = 92$]: $W = 1440$, $p = 0.006$, $r\_rb = 0.663$, [CI: −45.04, −6.007]).

**Psychophysical modeling.** Similar to Experiments 1A and 1B, the average staircase ending point at block ends across participants revealed a similar tendency to rate suppressed locations as darker than their non-suppressed counterpart (one-tailed t-test: $t_{(99)} = 1.892$, $p = 0.031$, $d = 0.189$, [CI: −∞, −0.573]). A subsequent rmANOVA with within subjects' factor Block (1-5) again failed to find a reliable blockwise effect (ggANOVA: $F_{(3.59,355.3)} = 0.295$, $p = 0.862$, $\eta^2 = 0.003$ [CI: 0.000, 0.014]). Furthermore, individual psychometric fits done to each participants data revealed that the PSE was again reliably below zero (one-tailed w-test: $W = 1898$, $p = 0.016$, $r\_rb = 0.624$, [CI: −∞, −0.61]; Fig. 4F). This indicates that the salience modulation observed in Experiment 1A and 1B cannot be attributed to the black and white search stimuli used.

**Computational modeling.** Similar to Experiments 1A and 1B, our model comparison consistently preferred the reduced capture model, outperforming both the disengagement and combined models based on AIC and BIC scores using the combined subject data. The winning capture model estimated 33% capture odds by a distractor at HP locations versus 80% at LP locations. Furthermore, an examination of the median curve values revealed a median RT on the no-capture curve of 714 ms and a median RT in the capture curve of 915 ms, indicating a true capture cost of 201 ms (calculated using our best performing capture model). This modeling approach was also applied to the individual participant data to get an idea of the consistency of the capture effect. Comparing individual AIC and BIC scores for the three models showed again that the capture model was consistently the best fitting model for the individual data, including being reliably better fitting when comparing BIC scores between the capture model and the second-place disengagement model

(AIC: F(2,198) = 12.08, $p < 0.001$, $\eta^2 = 0.109$ [CI: 0.028, 0.189]. BIC: F(2,198) = 377.0, $p < 0.001$, $\eta^2 = 0.792$ [CI: 0.742, 0.843]. Pairwise comparisons shown in Fig. 4G).

## General discussion

The brain's ability to filter irrelevant information in favor of task-relevant stimuli is a fundamental aspect of attention, crucial for efficient cognitive processing[1,3]. The current study demonstrates two important features of learned attentional suppression in response to spatial distractor regularities. First, items presented at frequently suppressed regions of space are perceived as less bright as compared to non-suppressed regions, suggesting a perceptual component of learned suppression. Second, and in line with predictions derived from the psychophysical findings, modeling of response times revealed that learned suppression primarily modulates capture probability at the suppressed region, rather than affecting disengagement speed. These findings suggest that learned suppression is *proactive* occurring at the earliest moments of competition calculations within the spatial attentional priority map.

### Perceptual consequences of learned suppression

Our psychophysical analyses of probe trials consistently demonstrated a bias to rate items presented at suppressed locations as less bright than those at non-suppressed locations. This effect was robust across both staircase ending points and PSE measures. Critically, these effects were consistent whether participants were tasked with selecting the brighter or darker probe, and whether the stimuli were colored or greyscale; ruling out the possibility of a general bias against selecting the suppressed location or deference to global priors. These results, which align with recent neuroimaging findings showing that ERPs associated with saliency judgments increase due to statistical learning[35,71], suggest that learning has a direct perceptual consequence that likely accounts for improved search efficiency within displays wherein a salient distractor appears at a HP distractor location.

This study adapted the psychophysical approach, commonly used in research on exogenous spatial attention[72,73] to investigate learned attentional effects. Learned suppression shares several characteristics with exogenous attention which has also been found to induce perceptual effects[73–76]. Previous work has highlighted the parallels between selection history (such as statistical learning) and exogenous attention, leading to the suggestion that both rely on the same attentional machinery[77]. Much like exogenous attention, learned attentional effects are imposed regardless of (and sometimes in opposition to) the observer's goals, are fast in comparison to endogenous attention, and require little effort to execute. Given these striking similarities between exogenous attention and selection history, and given the computational modeling results discussed below, one can ask: is the salience effect observed here simply a reflection of exogenous attention being attracted away from the HP distractor location? After all, it has been shown that deploying attention to one location in space results in a change in the appearance of objects at that location[73,74]. The question then becomes which comes first: the saliency effect or the priority effect? A reduction in saliency will certainly lead to a reduction in priority, which may then lead to a reduction in sensitivity, creating a feedback loop in which the initial cause is difficult to disentangle. While it is plausible that the salience effect is caused by a change in priority, rather than causing one, it is important to note that often experiments studying the role of attention on appearance demonstrate a change in sensitivity at attended locations, not a saliency modulation per se. As such, we find the former model more plausible, though our results cannot rule out the latter and more work is needed to this end.

While there is some precedent to our study showing the role of learning on perception[78], perceptual questions have traditionally been a greater interest to the field of predictive processing[79]. Work on prediction and perception has robustly demonstrated the influence of expectations on visual[80–82] as well as tactile perception[83,84]. Furthermore, plausible neural models have identified differential processing across cortical and laminar layers corresponding to the predictive influence on perception[85–87]. It is thus apt to ask whether the predictive mechanisms are one and the same as the learning mechanisms underlying our observed effect. A key difference between the statistical learning results and those from the field of predictive processing is the role of top-down guidance, as prediction effects are thought to be highly flexible, responding to trial-wise cues. Learning, on the other hand, is seen as slow and durable, requiring many trials of experience accumulation. As such, it remains an open question of interest whether the currently observed perceptual results of learning are one and the same as the more dynamic influences observed in predictive processing literature, and what role top-down guidance plays on distinguishing between the two.

### Learning proactively reduces competitiveness of stimuli at suppressed locations, resulting in reduced capture episodes

Our computational modeling of RTs favored a model of reduced capture rate as the primary outcome of learned suppression. This model outperformed alternatives that assumed consistent capture with faster disengagement or a combination of both mechanisms. These results suggest that learned suppression primarily affects the earliest stages of attentional processing by reducing the likelihood of initial capture episodes. Previous work seeking to mediate between capture and disengagement models of learned suppression have focused on eye tracking results[29,65,66]. Consistent with the current findings, eye tracking studies have shown that the largest effect of learned suppression is a reduction of the number of saccades made towards this suppressed region. Yet, these studies also revealed some small effects associated with an increased speed with which eyes were disengaged from a suppressed relative to an unsuppressed location. Because there can be attentional capture to a location in space without a subsequent overt eye movement[88], one cannot rely on eye movements alone to model covert attentional shifts. Here we used a method of capturing this covert variation by modeling the distributional properties of attentional behavior. While it remains likely that disengagement, as observed by previous eye movement studies, has some role in learned suppression, the current results demonstrate clearly that the most influential factor is a reduction in capture episodes.

Attention has successfully been modeled as a winner-take-all competition between priority signals in space, where attentional priority is organized in a spatial map with various priority ratings. According to the tripartite model of attention control (Theeuwes, 2025) selection is the result of interaction and competition among top-down goals, bottom-up saliency, and selection-history influences[9,20,27]. The current results show that selection history may affect initial capture by reducing the saliency of objects presented in suppressed regions, thereby decreasing their competition for neural representation[18].

### Integrating the current results with theories of learned suppression

These results align with the recently proposed synaptic theory of learned attention[32–34], which posits that the benefits of targets and distractors appearing at HP locations are mediated by increased/decreased synaptic connectivity in enhanced/suppressed regions of space. Particularly noteworthy is recent work demonstrating that a learned spatial bias can be decoded from task-irrelevant high-contrast perturbations presented during the interval before the search display is presented[32]. This decoding was interpreted as compelling evidence for a proactive synaptic mechanism, as it revealed a modulation of the baseline activity. The current results are consistent with these findings, and extend them into the perceptual domain, showing that the neural decoding effects reported by Duncan et al.[35] have a perceptual counterpart reflected by changes in saliency judgments. This suggests that learned suppression primarily operates by changing the bottom-up saliency signal of items in space.

The current results corroborate a host of behavioral[30,89–93] and neuroimaging[31–33,94,95] findings supporting the proactive nature of learned suppression. A key assumption of proactive suppression is that it exerts itself on the initial feedforward sweep of attention, and that it generalizes to new

stimuli presented in the suppressed region. The current results demonstrate evidence for both: that suppression affects the initial capture episode, indicating an influence on initial saliency calculations, and the spreading of suppression from the search task to the probe task similarly indicates a generalization of the learning effect across stimuli set and tasks.

The current findings are also consistent with the notion of incentive salience, highlighting how learning can alter the low-level representation of stimuli associated with reward[96,97]. This work has suggested that these changes occur at the fundamental level of stimuli processing indicating that learning can reshape how stimuli are encoded from the earliest stages of perception (for review, see)[98]. Statistical learning and reward learning are similar selection history mechanisms[77]. These findings agree with a host of neurophysiological findings that suggest their underlying mechanisms are similar[35,71,99,100], and suggest they jointly operate by modifying the low-level representations of stimuli.

However, the current findings leave open several major theoretical questions. A critical open question remains whether the current findings represent pure suppression of one location in space, or alternatively the enhancement of the opposite side of the screen. Questions of pure suppression versus enhancement are notoriously difficult to answer due to the need for a consistent neutral baseline during measurement. Furthermore, from the biological perspective, it is unclear how suppression and enhancement would arise from the interaction of neurons in the primary visual cortex[17,101]. One influential model of suppression is that of divisive normalization, which asserts that one finite attentional resource exists at the level of salience competition in the visual cortex[102–104]. The concept of a finite resource necessarily means that enhancement of one location requires suppression of somewhere else, providing a potential theoretical resolution to the question of whether learning leads to pure suppression or enhancement. Importantly, a direct prediction of this model would be that attentional enhancement should show the exact opposite pattern of perceptual salience modulation as observed here, as attentional enhancement should operate using the exact same mechanism: shifting the distribution of a finite attentional resource over space. While divisive normalization is a compelling theoretical framework within which to frame our learning effects, very little work has been done integrating statistical learning with this formal theory of neural computation, and so more work is needed to explore whether this theory does indeed provide a good account for how learning shifts attentional priority.

Beyond this critical theoretical question on the nature of attentional suppression, there remain other key open questions which cannot be answered by the current data. Among these perhaps the most pressing is the question of whether feature learning differs from spatial learning and whether the modeling approach described here would identify a similar proactive mechanism underlying the speeded rejection of colored distractors presented in a HP color. Feature learning is a popular field of study with many overlaps to spatial learning[105–109] and it remains a vibrant topic of debate whether feature learning shares the same cognitive mechanisms as spatial learning[110–113]. This is especially prescient in the context of the Signal Suppression hypothesis, which proposes that featural information can be proactively downweighed such that certain highly salient features no longer capture attention[105]. The cognitive modeling approach outlined here may offer a compelling tool to further compare the cognitive mechanisms of these two learning effects. Furthermore, it remains unclear whether the current results would share the same pattern if distractors were defined as darker than the background rather than lighter. It seems reasonable to presume that distractors appeared less bright at suppressed locations in this paradigm simply as a result of presenting bright shapes on a black background, and that reversing this contrast would result in a corresponding reversal in the pattern of results (i.e., participants would reliably rate shapes as brighter at suppressed locations if they were presented on a white background). The reason for this speculation is that statistical learning modulates relative salience, or the contrast between an item and its surrounding. As a result, any modulations via learning should result in items presented at suppressed areas appearing more similar to their surroundings

than elsewhere. However, this prediction remains purely theoretical and a topic for future investigations.

In summary, the current computational modeling approach, in conjunction with the psychophysical modeling results, provides strong evidence that learned suppression primarily operates by reducing the likelihood of initial attentional capture, rather than by facilitating faster disengagement. These results suggest that selection history modulates perceptual salience within the initial feedforward sweep of information processing such that stimuli at suppressed regions become less salient and hence less competitive within the spatial priority map. In combination with top-down and bottom-up influences this perceptual modulation then results in reduced capture by salient items at the suppressed location (Fig. 3A). Together, these findings provide a comprehensive account of how learned spatial suppression shapes attentional priorities and perceptual experiences

### Limitations
Because the current study was collected online, it remains to be seen how in person collection would affect the observed pattern of results. It is the opinion of the authors that with more experimental control, the effect would become stronger, necessitating less participants to overcome noise in the data.

### Code availability
The experimental code and analysis scripts are available online (https://osf.io/25f76/).

### Data availability
The data anonymized participant data are available online (https://osf.io/25f76/).

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

## Acknowledgements

J.T. was supported by a European Research Council (ERC) advanced grant [Grant #: 833029—LEARNATTEND] and an Nederlandse Organisatie voor Wetenschappelijk Onderzoek (NWO) SSH Open Competition Behavior and Education grant [Grant #: 406.21.GO.034]. The funders had no role in study design, data collection and analysis, decision to publish or preparation of the manuscript.

## Author contributions

Original idea by D.H.D., D.v.M., and J.T. Experiment programmed by D.H.D. Data collected by D.H.D. Data analyzed by D.H.D. Manuscript written by D.H.D., D.v.M., and J.T.

## Competing interests

The authors declare no competing interests
