## [Transparent Peer Review file · Communications Psychology]

Learning Alters Salience and Proactive Attentional Priority

Corresponding Author: Dr Dock Duncan

Version 0:

Decision Letter:

Dear Dr Duncan,

Thank you for your patience during the peer-review process. Your manuscript titled "Learning Alters Salience and Priority" has now been seen by 2 reviewers, and I include their comments at the end of this message. They find your work of interest but raised some important points. We are interested in the possibility of publishing your study in Communications Psychology, but would like to consider your responses to these concerns and assess a revised manuscript before we make a final decision on publication.

We therefore invite you to revise and resubmit your manuscript, along with a point-by-point response to the reviewers. Please highlight all changes in the manuscript text file.

Editorially, we consider it important that you address all methodological concerns with additional analysis and/or experimentation. Alternative explanations for the central results of the study should also be included.

Please ensure you follow our statistical guidelines when reporting statistics (<https://www.nature.com/commspsychol/submit/submission-guidelines#statistical-guidelines>). Please note in particular our requirements for the reporting and interpretation of null-results. Non-significant findings derived from null-hypotheses significance tests should be reported in full, but may not be interpreted. Where you interpret null results, this interpretation must be based on Bayes Factors or equivalence tests.

I am attaching an Editorial Requests Table that details critical reporting requirements for the revised manuscript. Please attend to each item and ensure your manuscript is fully compliant. If your revised manuscript is not aligned with these requests on major issues, such as those concerning statistics, it may be returned to you for further revisions without re-review.

Please submit the following items:

- Revised manuscript
- Point-by-point response to the referees' comments
- Cover letter (as a separate document)
- <https://www.nature.com/documents/nr-reporting-summary.pdf> Nature Research Reporting Summary
- Completed Editorial Request Table (attached).

via this link: Link Redacted .

Additional guidance is available in our style and formatting guide Communications Psychology formatting guide.

Best regards,

Troy Lui

Troy Lui, PhD
Associate Editor
Communications Psychology

REVIEWER EXPERTISE:

Reviewer #1: visual attention, distractor suppression

Reviewer #2: selection history, cognitive modelling

REVIEWER REPORTS:

Reviewer #1 (Remarks to the Author):

COMMSPSYCHOL-25-0551

Learning Alters Salience and Priority

In this original study, the authors examined the behavioral/perceptual relevance of spatial statistical learning in the context of visual search. For this purpose, in a series of experiments, the authors use an adapted version of an established additional singleton task with an imbalanced and defined distractor distribution together with a subjective brightness task. They overall find that the imbalanced spatial distribution of distractors across trials of the visual search task leads to altered processing of targets and distractors in the visual search task and reduced perceived brightness for stimuli presented at the high-probability distractor position. In addition, modelling of the reaction time distributions of responses during the visual search task seems to point towards reduced capture by high-probability distractors rather than faster disengagement of attention from high-probability distractors.

The research question is timely and quite relevant to the field as well as beyond. The experimental design is straightforward, and potential alternative explanations to the initial findings were addressed in a pre-registered control of two experiments. The implemented analyses are sound and elaborate, and followed the registration for the two additional experiments. The results reported for the set of three different experiments point towards a consistent and reliable effect of statistical learning on behavior/perception.

The manuscript is overall well-written and concise, and I have a few suggestions that may need to be addressed.

The psychophysical modelling approach is based on the idea that bimodal reaction time distributions may arise from differential selection sequences in each trial, i.e., responses vary as targets may be directly selected or targets may only be selected after a salient distractor captured attention initially. The modelled distributions underlying the bimodal RT distributions may thus contain information on attentional dynamics related to the shifts of attention, and it may be of interest to readers to report some descriptives of the underlying distributions. I.e., what was the median reaction time for the modelled distributions of capture and direct target trials?

The conducted analyses of the data clearly point towards differential processing of stimuli at the high probability distractor position. These changes are interpreted as a suppression of the spatial location of the high-probability distractor position. However, the underlying findings are based on relative differences between conditions derived from the responses to target

stimuli presented at various locations. While the data may be explained by a suppression of the high probability distractor location, it may alternatively also stem from an amplification or upweighting of the other locations. In fact, the authors mention in the introduction that the "refinement of the priority map emphasizes previously relevant locations and de-emphasizes previously distracting locations." Especially with regards to a very prominent but also debated theory claiming proactive suppression as a key mechanism for neural processing in visual search, the manuscript may benefit from a more nuanced discussion with regards to underlying amplification/suppression effects.

The current findings are summarized as "selection history modulates perceptual salience within the initial feedforward sweep of information processing such that stimuli at suppressed regions become less salient and hence less competitive within the spatial priority map." Despite the framing of solely suppression being at work (see point above), I find the evidence and the modelling quite compelling, contrasting faster re-engagement and reduced capture. I'm, however, struggling with bringing these results together with some previous work by some of the authors. How do the here reported data relate to ingroup work that reports data rather, "suggesting initial spatial selection followed by suppression" in a statistical learning paradigm?

Huang, C., van Moorselaar, D., Foster, J., Donk, M. & Theeuwes, J. Neural mechanisms of learned suppression uncovered by probing the hidden attentional priority map. *Elife* 13, (2025).

MINOR:

The authors described a post-experimental color matching task (line 288 onwards). Are the results reported somewhere? If not, it may be of interest whether all participants were able to do this technical test or whether participants were excluded for not meeting some criteria of this technical test.

On several occasions in the manuscript, the wrong figure is referenced in the text, e.g., figure 3 is referenced although it should be figure 2 (line 448, for instance). Please check.

For clarity, the manuscript may benefit from an exemplary display of the stimulus display of experiment 2 (similar to or as an add-on to Figure 1).

Reviewer #2 (Remarks to the Author):

The manuscript describes two experiments and modeling results to provide evidence for the claim that learned distractor suppression operates by reducing capture rather than facilitating disengagement from the distractor. In experiment 1 (a+b), a dark distractor location is learned, subsequently this objects at location are shown to be perceived as less bright than at other locations. In experiment 2, color-based distractors are used. Both psychophysical and modeling results point in the same direction as in experiment 1, although not as strongly.

The paper is generally well written and the results are mostly convincing. Detailed comments and suggested (major) revisions below:

Intro line 57-91: Distractor suppression can also be feature-based, please also cite recent works on feature-based Distractor suppression, e.g.

Aylin A. Hanne, Sizhu Han, Anna Schubö; Neural Evidence for Feature-based Distractor Inhibition. *J Cogn Neurosci* 2025; 37 (6): 1053–1071. doi: <https://doi.org/10.1162/>

I would also like to encourage the authors to speculate on what kind of saliency (?) effects to expect if their experiment was repeated with distractors having featural rather than spatial regularities.

Methods line 185 "Finally, if a participant selected the wrong item during the saliency probe on more than 45% of trials, they were excluded and replaced (21 participants replaced)." -> Does this not bias the result in favor of your hypothesis, since the "wrong item" is presumably defined relative to the task?

207 "We chose to use consistently darker distractors than targets to avoid chromatic adaptation" OK, but what happens if the distractor would be brighter? Could people still adapt? This should be tested & reported. I appreciate the chromatic adaptation argument and the data presented in experiment 2, but a brighter distractor experiment would strengthen the claims of the paper.

245 . Critically, the brightness of the item on nonstandard side .. which side is that? The one with or without the distractor?

line 259-270: this stepsize halving approach in the staircase is very similar to the standard interval bisection method uses in root finding (see e.g. https://en.wikipedia.org/wiki/Bisection_method). If that was stated at the beginning of the paragraph, I would have found it easier to follow

Fig 2, F+G bottom panels: the resolution of the figure is so low that I cannot discern what I am supposed to see (line 340)

line 355: "A comparison of AIC scores across curve fits..." using AIC for model comparison offer overfits, please use BIC

(even though I expect the same results given your fits). Regarding use of ex-Gaussians: I agree with the choice for pragmatic reasons, for a theoretically more well-founded construction of reaction time distributions, see <https://escholarship.org/uc/item/1nj6m2n7>

lines 556ff: the pdf is faulty, but I got the gist of experiment 2

Fig 4E: the AIC/BIC scores look almost identical to those in fig. 3, please check.

line 2520, general discussion: "... is fast in comparison..." -> are fast in comparison

Version 1:

Decision Letter:

Dear Dr Duncan,

Your manuscript titled "Learning Alters Salience and Priority" has now been seen by our reviewers, whose comments appear below. In light of their advice I am delighted to say that we are happy, in principle, to publish a suitably revised version in Communications Psychology.

We therefore invite you to revise your paper one last time to address the remaining concerns of our reviewers and a list of editorial requests. At the same time we ask that you edit your manuscript to comply with our format requirements and to maximise the accessibility and therefore the impact of your work.

EDITORIAL REQUESTS:

SUBMISSION INFORMATION:

OPEN ACCESS:

Communications Psychology is a fully open access journal. Articles are made freely accessible on publication. For further information about article processing charges, open access funding, and advice and support from Nature Research, please

visit <https://www.nature.com/commpsychol/open-access>

* **DATA AVAILABILITY:**

Link Redacted

Best regards,

Troy Lui

Troy Lui, PhD
Associate Editor
Communications Psychology

REVIEWERS' COMMENTS:

Reviewer #1 (Remarks to the Author):

The authors have fully addressed all the points raised in my previous review. I have no further comments and recommend the manuscript for publication in its current form.

Reviewer #2 (Remarks to the Author):

Dear Authors,

thank you for your rebuttal and revision. I have only two minor comments:

- Supplementary Figure 2: is the horizontal axis unit really ms?

- the reference "Meibodi, N., Schubö, A., & Endres, D. M. (2022). Sensorimotor processes are not a source of much noise: Sensory-motor and decision components of reaction times. 44(44)" should be "Meibodi, N., Schubö, A., & Endres, D. M. (2022). Sensorimotor processes are not a source of much noise: Sensory-motor and decision components of reaction times. Proceedings of the Annual Meeting of the Cognitive Science Society, 44."

best,

the reviewer

We thank both reviewers for their positive assessment of our manuscript and also for their very constructive feedback which we believe has led to an improved manuscript. We address each of their points in order below, including pasting the in-text revisions which their comment brought about. The reviewers' original comments are written in black. Our responses to their comments are written in blue, and the in text changes are written in red.

In addition to changes brought on by the reviewers' comments, we also modified our statistics to adhere to the journal guidelines. In no cases did this lead to any changes to our previous pattern of results. We have added the following text in the methods section addressing these statistical methods:

In text revision. Page 9:

Statistical Tests

Our analyses will rely on simple repeated measure analyses of variance (rmANOVA) or t-tests. Ninety-five percent confidence intervals (CI) reported for t-tests indicate the interval of the reaction time difference between conditions. In all cases assumptions of sphericity (for ANOVAs) or regularity (for t-tests) will be conducted and, if violated, the proper corrected statistical test will be performed. In the case of rmANOVAs, Greenhouse-Geisser corrected ANOVAs (ggANOVA) will be conducted, and for t-tests, Wilcoxon signed-rank test (w-test) will be performed instead. As can be seen in the reaction time modelling data, (Figure 3D) our results generally did not follow a normal distribution, thus motivating a nonparametric test. Rather than normalizing the data (e.g., via log transformation), we chose to preserve reaction time information and defer to appropriate non-parametric tests.

Reviewer #1 (Remarks to the Author):

COMMSPSYCHOL-25-0551

Learning Alters Saliency and Priority

In this original study, the authors examined the behavioral/perceptual relevance of spatial statistical learning in the context of visual search. For this purpose, in a series of experiments, the authors use an adapted version of an established additional singleton task with an imbalanced and defined distractor distribution together with a subjective brightness task. They overall find that the imbalanced spatial distribution of distractors across trials of the visual search task leads to altered processing of targets and distractors in the visual search task and reduced perceived brightness for stimuli presented at the high-probability distractor position. In addition, modelling of the reaction time distributions of responses during the visual search task seems to point towards reduced capture by high-probability distractors rather than faster disengagement of attention from high-probability distractors.

The research question is timely and quite relevant to the field as well as beyond. The experimental design is straightforward, and potential alternative explanations to the initial

findings were addressed in a pre-registered control of two experiments. The implemented analyses are sound and elaborate, and followed the registration for the two additional experiments. The results reported for the set of three different experiments point towards a consistent and reliable effect of statistical learning on behavior/perception.

The manuscript is overall well-written and concise, and I have a few suggestions that may need to be addressed.

We thank the reviewer for their positive and comprehensive appraisal of our work and for their time in reviewing and providing comments which we believe have resulted in an improved manuscript.

1. The psychophysical modelling approach is based on the idea that bimodal reaction time distributions may arise from differential selection sequences in each trial, i.e., responses vary as targets may be directly selected or targets may only be selected after a salient distractor captured attention initially. The modelled distributions underlying the bimodal RT distributions may thus contain information on attentional dynamics related to the shifts of attention, and it may be of interest to readers to report some descriptives of the underlying distributions. I.e., what was the median reaction time for the modelled distributions of capture and direct target trials?

This is an excellent point that we had not previously considered. Upon examining the median RT for capture and no capture trials (using the best performing capture model) we found that delays due to distraction were around 215 milliseconds. Notably, this distraction cost is roughly twice the difference between no distractor trials and low-probability distractor trial reaction times. This pattern makes sense under our model which found that capture only occurs on approximately 70% of the distractor present trials, the aggregate average reaction times blend the temporal cost of actual capture events with a subset of trials in which no capture by the distractor occurred. Thus, the aggregate measures underestimate the true attentional cost when capture does occur. We now include this important information in the manuscript.

In text revisions. Page 14:

... Additionally, our modelled response time curves provided information on the median value for capture and non-capture trials, offering a refined estimate of the temporal cost of singleton capture. Both models used a no-capture curve with a median reaction time value at 714 ms. The reduced capture model further estimated a median reaction time on capture trials of 929 ms, yielding a distractor cost of 215 ms. Note that this distractor cost is roughly twice the difference between distractor absent and low-probability distractor trial reaction times, indicating that the presence of no-capture trials in the aggregate reaction times obscures the true temporal cost of visual distraction.

2. The conducted analyses of the data clearly point towards differential processing of stimuli

at the high probability distractor position. These changes are interpreted as a suppression of the spatial location of the high-probability distractor position. However, the underlying findings are based on relative differences between conditions derived from the responses to target stimuli presented at various locations. While the data may be explained by a suppression of the high probability distractor location, it may alternatively also stem from an amplification or upweighting of the other locations. In fact, the authors mention in the introduction that the “refinement of the priority map emphasizes previously relevant locations and de-emphasizes previously distracting locations.” Especially with regards to a very prominent but also debated theory claiming proactive suppression as a key mechanism for neural processing in visual search, the manuscript may benefit from a more nuanced discussion with regards to underlying amplification/suppression effects.

We agree that this is a complex and important debate. Suppression and enhancement are indeed difficult to disentangle experimentally, as it is challenging to demonstrate that suppression in one location did not result in enhancement elsewhere. In fact, we are more partial to the divisive normalization approach to this question, which suggests parallel suppression in the visual system will always pair enhancement with suppression. Essentially, the divisive normalization account is a finite resource model where if attention is not going to one location, it necessarily must be going to another. We have now expanded our discussion of this theoretical framework in the manuscript.

In text revisions. Page 22:

However, the current findings leave open several major theoretical questions. A critical open question remains whether the current findings represent pure suppression of one location in space, or alternatively the enhancement of the opposite side of the screen. Questions of pure suppression versus enhancement are notoriously difficult to answer due to the need for a consistent neutral baseline during measurement. Furthermore, from the biological perspective, it is unclear how suppression and enhancement would arise from the interaction of neurons in the primary visual cortex (Chelazzi et al., 2019; Klink et al., 2023). One influential model of suppression is that of divisive normalization, which asserts that one finite attentional resource exists at the level of salience competition in the visual cortex (Carandini & Heeger, 2012; Heeger, 1992; Reynolds & Heeger, 2009). The concept of a finite resource necessarily means that enhancement of one location requires suppression of other locations in space, providing a potential theoretical resolution to the question of whether learning leads to pure suppression or enhancement. Importantly, a direct prediction of this model would be that attentional enhancement should show the exact opposite pattern of perceptual salience modulation as observed here, as attentional enhancement should operate using the exact same mechanism: shifting the distribution of a finite attentional resource over space. While divisive normalization is a compelling theoretical framework within which to frame our learning effects, very little work has been done integrating statistical learning with this formal theory of neural computation, and so more work is needed to explore whether this theory does indeed provide a good account for how learning shifts attentional priority.

3. The current findings are summarized as "selection history modulates perceptual salience within the initial feedforward sweep of information processing such that stimuli at suppressed regions become less salient and hence less competitive within the spatial priority map." Despite the framing of solely suppression being at work (see point above), I find the evidence and the modelling quite compelling, contrasting faster re-engagement and reduced capture. I'm, however, struggling with bringing these results together with some previous work by some of the authors. How do the here reported data relate to ingroup work that reports data rather, "suggesting initial spatial selection followed by suppression" in a statistical learning paradigm?

Huang, C., van Moorselaar, D., Foster, J., Donk, M. & Theeuwes, J. Neural mechanisms of learned suppression uncovered by probing the hidden attentional priority map. *Elife* 13, (2025).

The reviewer is correct in pointing out that the current results seem at odds with those from our previous project using multivariate decoding of EEG results. We have spent quite some time considering why this one project seemed to stand alone in supporting a reactive view of statistical learning. Our interpretation now hinges on two points: firstly, the method using a concurrent working memory and search task while probing working memory maintenance using a ping was quite novel and it is unclear exactly what neural processes this ping was uncovering. Specifically observing how learning influences the EEG measures of working memory is a rather circuitous way of approaching the question on neural mechanisms of statistical learning. It is possible that learned suppression may lead to reduced interference at the memory location, for instance, or that decreased attentional priority at this location led to better neural representations at this location (it is, for instance, known that the ocular system avoids locations held in spatial working memory as some sort of natural shield for working memory interference. By proactively reducing attentional priority at this location, we may have decreased the need for such shielding, and this may have led to better decoding of space when the ping onset.). Secondly, it has been shown in the past that learning consists of both proactive and reactive components, and the main purpose of research such as what we report here isn't to support either a totally proactive or a totally reactive model of statistical learning, but to give evidence for which is more influential (this is why in this paper we are careful to mention that reactive processes are still a known contributor to learned suppression, and our modelling results simply highlight that proactive processes are more influential than these reactive processes). It is possible that the specific methods used in our eLife paper may have been optimized for picking up on these reactive influences while minimizing the neural influence of proactive processes. However again this is pure speculation, and we agree with the reviewer that this is a puzzling question that is deserving of further research.

MINOR:

4. The authors described a post-experimental color matching task (line 288 onwards). Are the

results reported somewhere? If not, it may be of interest whether all participants were able to do this technical test or whether participants were excluded for not meeting some criteria of this technical test.

The color matching task was used as an exclusion criterion – if participants were unable to match the colors on both sides of the screen, then their data was simply excluded. This is addressed in the participants section in the methods.

5. On several occasions in the manuscript, the wrong figure is referenced in the text, e.g., figure 3 is referenced although it should be figure 2 (line 448, for instance). Please check.

Thank you for catching this oversight. We have carefully gone over the manuscript to ensure that the correct figure is consistently referenced.

6. For clarity, the manuscript may benefit from an exemplary display of the stimulus display of experiment 2 (similar to or as an add-on to Figure 1).

This has been added as requested

in text revision. Page 17, Figure 4:

Reviewer #2 (Remarks to the Author):

The manuscript describes two experiments and modeling results to provide evidence for the claim that learned distractor suppression operates by reducing capture rather than facilitating disengagement from the distractor. In experiment 1 (a+b), a dark distractor location is learned, subsequently this objects at location are shown to be perceived as less bright than at other locations. In experiment 2, color-based distractors are used. Both psychophysical and modeling results point in the same direction as in experiment 1, although not as strongly. The paper is generally well written and the results are mostly convincing. Detailed comments and suggested (major) revisions below:

We thank the reviewer for their constructive feedback and apologize that the image quality was no up to par in the previous submission. We hope that this has been fixed in the new submission (at least we will check the built PDF very carefully).

1. Intro line 57-91: Distractor suppression can also be feature-based, please also cite recent works on feature-based Distractor suppression, e.g.

Aylin A. Hanne, Sizhu Han, Anna Schubö; Neural Evidence for Feature-based Distractor Inhibition. *J Cogn Neurosci* 2025; 37 (6): 1053–1071. doi: <https://doi.org/10.1162/>

The reviewer is correct in that we have neglected to refer to several important feature based distractor learning studies. We have now endeavored to address these studies in greater detail in the discussion section of the text.

In text revisions. Page 22:

Beyond this critical theoretical question on the nature of attentional suppression, there remain other key open questions which cannot be answered by the current data. Among these perhaps the most pressing is the question of whether feature learning differs from spatial learning and whether the novel modelling approach described here would identify a similar proactive mechanism underlying the speeded rejection of colored distractors presented in a high-probability color. Feature learning is a popular field of study with many overlaps to spatial learning (Gaspelin et al., 2025; Hanne et al., 2025; Stilwell et al., 2019; van Moorselaar et al., 2020; S. Wang et al., 2023) and it remains a vibrant topic of debate whether feature learning shares the same cognitive mechanisms as spatial learning (Hanne et al., 2023; Liesefeld & Müller, 2019; Sauter et al., 2018, 2019). This is especially prescient in the context of the Signal Suppression hypothesis, which proposes that featural information can be proactively downweighed such that certain highly salient features no longer capture attention (Gaspelin et al., 2025). The cognitive modelling approach outlined here may offer a compelling tool for further compare the cognitive mechanisms of these two learning effects.
...

2. I would also like to encourage the authors to speculate on what kind of saliency (?) effects

to expect if their experiment was repeated with distractors having featural rather than spatial regularities.

In line with the above comment, we have now also include a brief discussion on exactly this question. We have specifically attempted to frame the current results in the context of the dimension weighting and the signals suppression hypotheses in an attempt to bring our current results in line with the literature

In text revisions:

(see revisions listed in Reviewer 2, comment 1)

3. Methods line 185 "Finally, if a participant selected the wrong item during the saliency probe on more than 45% of trials, they were excluded and replaced (21 participants replaced)." -> Does this not bias the result in favor of your hypothesis, since the "wrong item" is presumably defined relative to the task?

This is an interesting question from the reviewer, the reason we chose to use this exclusion criteria is that some participants during the experiment simply pressed the same key on every single saliency probe trial. These criteria allowed us to exclude these participants easily, but we hadn't considered exactly how this may influence our results. Upon considering the reviewers concerns, we believe that if anything this may have biased our data away from our hypothesis (as it excluded participants whose point of subjective equality was constantly drifting towards an extreme value), and in general we do not believe that a participant could get less than a 45% accuracy unless they were pressing random numbers. Think for example the situation where no learning effect has occurred, and the staircase center point is correctly at the zero position. Then if the participant is doing the task correctly, they will have a 100% accuracy. If, on the other hand, the staircase middle point (the point around which it is vacillating) is slightly positive. Then on 50% of the trials the participant will select the truly brighter shape, and on 50% of the trials they will select what is in all truth the darker shape. Thus, if they are performing the task correctly, and if their point of subjective equality is shifted, they will still select the 'correct' shape 50% of the time. The only situation they will select the 'incorrect' shape multiple times in a row is if their point of subjective equality has shifted to become more extreme (i.e. further from true equality). Even then, to get an accuracy of less than 45% would imply a strong and concerted bias across the experiment towards an extreme value. So, for these reasons, we do not believe that this step would have had a significant effect on our data beyond filtering out low quality participants.

4. 207 "We chose to use consistently darker distractors than targets to avoid chromatic adaptation" OK, but what happens if the distractor would be brighter? Could people still adapt? This should be tested & reported. I appreciate the chromatic adaptation argument and

the data presented in experiment 2, but a brighter distractor experiment would strengthen the claims of the paper.

We appreciate the reviewer's comment and agree that this is a very interesting question that deserves investigation. However, after running three experiments we also realize that this particular experimental design calls for very large participant numbers (experiment 1 in retrospect was very underpowered), we believe a sample size above 100 is probably justified, and unfortunately, that comes with a price tag of over 1,300 euros after participant and Prolific fees. For this project we sadly do not have the finances to run such an expensive follow-up experiment. But nevertheless this is a very good question and so we have highlighted this as an open question in our discussion to encourage future lines of research, and also to draw attention to the nature of relative salience that is investigated in this paper.

In text revisions. Pages 22-23:

...Furthermore, it remains unclear whether the current results would share the same pattern if distractors were defined as darker than the background rather than lighter. It seems reasonable to presume that distractors appeared less bright at suppressed locations in this paradigm simply as a result of presenting bright shapes on a black background, and that reversing this contrast would result in a corresponding reversal in the pattern of results (i.e. participants would reliably rate shapes as brighter at suppressed locations if they were presented on a white background). The reason for this speculation is that statistical learning modulates relative salience, or the contrast between an item and its surrounding. As a result, any modulations via learning should result in items presented at suppressed areas appearing more similar to their surroundings than elsewhere. However, this prediction remains purely theoretical and a topic for future investigations.

5. 245 . Critically, the brightness of the item on nonstandard side .. which side is that? The one with or without the distractor?

We agree that this phrasing was unclear and have changed it accordingly. We intended to reference the nonstandard colored shape (the one that could vary in its color value) which appeared on the opposite side of the screen as the standard shape (which was a set color value). The text has been revised to clarify this distinction.

In text revisions. Page 7:

The standard colored shape could be presented on either the right or left side of the screen, randomly selected on each trial. Critically, the brightness of the nonstandard color item present on the opposite side was controlled by way of an adaptive staircase procedure meant to home in on the point of subjective equality by identifying the location where participants were equally likely to select the standard item or the staircased item (50% likelihood point)...

6. line 259-270: this stepsize halving approach in the staircase is very similar to the standard interval bisection method used in root finding (see

e.g. https://en.wikipedia.org/wiki/Bisection_method). If that was stated at the beginning of the paragraph, I would have found it easier to follow

We now explicitly reference this method at the beginning of this part of the methods section to prime reviewers to this prior knowledge.

In text revisions. Page 7:

... The staircasing method was based on the interval bisection method used in root finding, and was essentially a simplified version of the ‘parameter estimation by sequential testing’ (PEST) method (Lieberman & Pentland, 1982; Taylor & Creelman, 1967)...

7. Fig 2, F+G bottom panels: the resolution of the figure is so low that I cannot discern what I am supposed to see (line 340)

We apologize for our oversight when confirming the pdf of our manuscript – it seems we did not do a proper job of ensuring that figures in the pdf were rendered appropriately, resulting in low resolution images in the version shared with reviewers. We have (hopefully) fixed this in the revised version of the text.

8. line 355: "A comparison of AIC scores across curve fits..." using AIC for model comparison often overfits, please use BIC (even though I expect the same results given your fits). Regarding use of ex-Gaussians: I agree with the choice for pragmatic reasons, for a theoretically more well-founded construction of reaction time distributions, see <https://escholarship.org/uc/item/1nj6m2n7>

We are happy to switch over to BIC for this model fit comparison though, as you have noted, this didn't change the pattern of results notably. We are also happy to add the suggested literature as a reference in our note about the recent coherence in the literature that has found exgaussian/inverse gaussian distributions are apt curves for capturing response time distributions. (Note that in the reference we chose not to specifically single out papers using inverse gaussian instead of exgaussian curve fits as the inverse gaussian is essentially just a simpler version of the exgaussian curve which is more restricted on its midpoint due to having one less variable for skew).

In text revisions. Page 10:

... A comparison of BIC scores across curve fits clearly identified the ex-gaussian distribution as the best fit to our RT data, and so we proceeded to fit our distractor present trials using ex-gaussian curves (note that this accords with recent work modelling RT distributions; Marmolejo-Ramos et al., 2023; Meibodi et al., 2022, 2024; Palmer et al., 2011)...

9. lines 556ff: the pdf is faulty, but I got the gist of experiment 2

Related to our response to question 7, we apologize that we did not properly check that our manuscript was uploaded correctly in the first draft. That has (hopefully) now been fixed in the second draft.

10. Fig 4E: the AIC/BIC scores look almost identical to those in fig. 3, please check.

We believe that the main issue here is in the scaling of these figures – because we used a very zoomed out scale, the figures appear very similar even though they are actually quite different. The reason we chose to use these scales was so that our AIC and BIC scores would use the same axis limits, but we realize now upon further consideration that this is not necessary as they report different statistics. As such we have adjusted the y-axis scales to better allow the data to be compared. We believe now the differences between Figure 3 and 4 should be more apparent (though the pattern of results is similar between these two figures).

11. line 2520, general discussion: "... is fast in comparison..." -> are fast in comparison

This has been fixed. Thank you for spotting the error